# Seq2SQL: Generating Structured Queries from Natural Language using Reinforcement Learning

## Abstract

Relational databases store a significant amount of the worlds data. However, accessing this data currently requires users to understand a query language such as SQL. We propose Seq2SQL, a deep neural network for translating natural language questions to corresponding SQL queries. Our model uses rewards from in-the-loop query execution over the database to learn a policy to generate the query, which contains unordered parts that are less suitable for optimization via cross entropy loss. Moreover, Seq2SQL leverages the structure of SQL to prune the space of generated queries and significantly simplify the generation problem. In addition to the model, we release WikiSQL, a dataset of 80654 hand-annotated examples of questions and SQL queries distributed across 24241 tables from Wikipedia that is an order of magnitude larger than comparable datasets. By applying policy-based reinforcement learning with a query execution environment to WikiSQL, Seq2SQL outperforms a state-of-the-art semantic parser, improving execution accuracy from 35.9% to 59.4% and logical form accuracy from 23.4% to 48.3%.

## 1 Introduction

Relational databases store a vast amount of today's information and provide the foundation of applications such as medical records (Hillestad et al., 2005), financial markets (Beck et al., 2000), and customer relations management (Ngai et al., 2009). However, accessing relational databases requires an understanding of query languages such as SQL, which, while powerful, is difficult to master. Natural language interfaces (NLI), a research area at the intersection of natural language processing and human-computer interactions, seeks to provide means for humans to interact with computers through the use of natural language (Androutsopoulos et al., 1995). We investigate one particular aspect of NLI applied to relational databases: translating natural language questions to SQL queries.

Our main contributions in this work are two-fold. First, we introduce Seq2SQL, a deep neural network for translating natural language questions to corresponding SQL queries. Seq2SQL, shown in Figure 1, consists of three components that leverage the structure of SQL to prune the output space of generated queries. Moreover, it uses policy-based reinforcement learning (RL) to generate the conditions of the query, which are unsuitable for optimization using cross entropy loss due to their unordered nature. We train Seq2SQL using a mixed objective, combining cross entropy losses and RL rewards from in-the-loop query execution on a database. These characteristics allow Seq2SQL to achieve state-of-the-art results on query generation.

Next, we release WikiSQL, a corpus of 80654 hand-annotated instances of natural language questions, SQL queries, and SQL tables extracted from 24241 HTML tables from Wikipedia. WikiSQL is an order of magnitude larger than previous semantic parsing datasets that provide logical forms along with natural language utterances. We release the tables used in WikiSQL both in raw JSON format as well as in the form of a SQL database. Along with WikiSQL, we release a query execution engine for the database used for in-the-loop query execution to learn the policy. On WikiSQL, Seq2SQL outperforms a previously state-of-the-art semantic parsing model by Dong & Lapata (2016), which obtains 35.9% execution accuracy, as well as an augmented pointer network baseline, which obtains 53.3% execution accuracy. By leveraging the inherent structure of

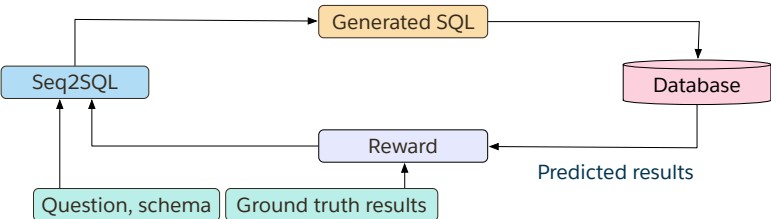

Figure 1: Seq2SQL takes as input a question and the columns of a table. It generates the corresponding SQL query, which, during training, is executed against a database. The result of the execution is utilized as the reward to train the reinforcement learning algorithm.

Table: CFLDraft

| Pick # | CFL Team | Player | Position | College |
|--------|----------|--------|----------|---------|
| 27 | Hamilton Tiger-Cats | Connor Healy | DB | Wilfrid Laurier |
| 28 | Calgary Stampeders | Anthony Forgone | OL | York |
| 29 | Ottawa Renegades | L.P. Ladouceur | DT | California |
| 30 | Toronto Argonauts | Frank Hoffman | DL | York |
| ... | ... | ... | ... | ... |

Question:
How many CFL teams are from York College?

SQL:
`SELECT COUNT CFL Team FROM CFLDraft WHERE College = "York"`

Result:
2

Figure 2: An example in WikiSQL. The inputs consist of a table and a question. The outputs consist of a ground truth SQL query and the corresponding result from execution.

SQL queries and applying policy gradient methods using reward signals from live query execution, Seq2SQL achieves state-of-the-art performance on WikiSQL, obtaining 59.4% execution accuracy.

## 2 MODEL

The WikiSQL task is to generate a SQL query from a natural language question and table schema. Our baseline model is the attentional sequence to sequence neural semantic parser proposed by Dong & Lapata (2016) that achieves state-of-the-art performance on a host of semantic parsing datasets without using hand-engineered grammar. However, the output space of the softmax in their Seq2Seq model is unnecessarily large for this task. In particular, we can limit the output space of the generated sequence to the union of the table schema, question utterance, and SQL key words. The resulting model is similar to a pointer network (Vinyals et al., 2015) with augmented inputs. We first describe the augmented pointer network model, then address its limitations in our definition of Seq2SQL, particularly with respect to generating unordered query conditions.

### 2.1 AUGMENTED POINTER NETWORK

The augmented pointer network generates the SQL query token-by-token by selecting from an input sequence. In our case, the input sequence is the concatenation of the column names, required for the selection column and the condition columns of the query, the question, required for the conditions of the query, and the limited vocabulary of the SQL language such as SELECT, COUNT etc. In the example shown in Figure 2, the column name tokens consist of "Pick", "#", "CFL", "Team" etc.; the question tokens consist of "How", "many", "CFL", "teams" etc.; the SQL tokens consist of SELECT, WHERE, COUNT, MIN, MAX etc. With this augmented input sequence, the pointer network can produce the SQL query by selecting exclusively from the input.

Suppose we have a list of $N$ table columns and a question such as in Figure 2, and want to produce the corresponding SQL query. Let $x_j^{\text{c}} = [x_{j,1}^{\text{c}}, x_{j,2}^{\text{c}}, ... x_{j,T_j}^{\text{c}}]$ denote the sequence of words in the name of the $j$th column, where $x_{j,i}^{\text{c}}$ represents the $i$th word in the $j$th column and $T_j$ represents the total number of words in the $j$th column. Similarly, let $x^{\text{q}}$ and $x^{\text{s}}$ respectively denote the sequence of words in the question and the set of unique words in the SQL vocabulary.

We define the input sequence $x$ as the concatenation of all the column names, the question, and the SQL vocabulary:

$$x = [\texttt{<col>}; x_1^{\text{c}}; x_2^{\text{c}}; ...; x_N^{\text{c}}; \texttt{<sql>}; x^{\text{s}}; \texttt{<question>}; x^{\text{q}}] \tag{1}$$

where $[a; b]$ denotes the concatenation between the sequences $a$ and $b$ and we add sentinel tokens between neighbouring sequences to demarcate the boundaries.

The network first encodes $x$ using a two-layer, bidirectional Long Short-Term Memory network (Hochreiter & Schmidhuber, 1997). The input to the encoder are the embeddings corresponding to words in the input sequence. We denote the output of the encoder by $h^{\mathrm{enc}}$, where $h_t^{\mathrm{enc}}$ is the state of the encoder corresponding to the $t^{\mathrm{th}}$ word in the input sequence. For brevity, we do not write out the LSTM equations, which are described by Hochreiter & Schmidhuber (1997). We then apply a pointer network similar to that proposed by Vinyals et al. (2015) to the input encodings $h^{\mathrm{enc}}$.

The decoder network uses a two layer, unidirectional LSTM. During each decoder step $s$, the decoder LSTM takes as input $y_{s-1}$, the query token generated during the previous decoding step, and outputs the state $g_s$. Next, the decoder produces a scalar attention score $\alpha_{s,t}^{\mathrm{ptr}}$ for each position $t$ of the input sequence:

$$\alpha_{s,t}^{\mathrm{ptr}} = W^{\mathrm{ptr}}\tanh\left(U^{\mathrm{ptr}}g_s + V^{\mathrm{ptr}}h_t\right) \tag{2}$$

We choose the input token with the highest score as the next token of the generated SQL query, $y_s = \mathrm{argmax}(\alpha_s^{\mathrm{ptr}})$.

## 2.2 Seq2SQL

While the augmented pointer network can solve the SQL generation problem, it does not leverage the structure inherent in SQL. Typically, a SQL query such as that shown in Figure 3 consists of three components. The first component is the aggregation operator, in this case COUNT, which produces a summary of the rows selected by the query. Alternatively the query may request no summary statistics, in which case an aggregation operator is not provided. The second component is the SELECT column(s), in this case Engine, which identifies the column(s) that are to be included in the returned results. The third

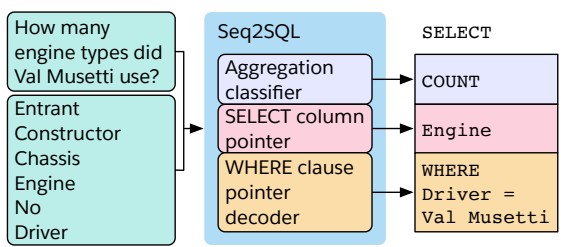

Figure 3: The Seq2SQL model has three components, corresponding to the three parts of a SQL query (right). The input to the model are the question (top left) and the table column names (bottom left).

component is the WHERE clause of the query, in this case WHERE Driver = Val Musetti, which contains conditions by which to filter the rows. Here, we keep rows in which the driver is "Val Musetti".

Seq2SQL, as shown in Figure 3, has three parts that correspond to the aggregation operator, the SELECT column, and the WHERE clause. First, the network classifies an aggregation operation for the query, with the addition of a null operation that corresponds to no aggregation. Next, the network points to a column in the input table corresponding to the SELECT column. Finally, the network generates the conditions for the query using a pointer network. The first two components are supervised using cross entropy loss, whereas the third generation component is trained using policy gradient to address the unordered nature of query conditions (we explain this in the subsequent **WHERE Clause** section). Utilizing the structure of SQL allows Seq2SQL to further prune the output space of queries, which leads to higher performance than Seq2Seq and the augmented pointer network.

**Aggregation Operation.** The aggregation operation depends on the question. For the example shown in Figure 3, the correct operator is COUNT because the question asks for "How many". To compute the aggregation operation, we first compute the scalar attention score, $\alpha_t^{\mathrm{inp}} = W^{\mathrm{inp}}h_t^{\mathrm{enc}}$, for each $t$th token in the input sequence. We normalize the vector of scores $\alpha^{\mathrm{inp}} = [\alpha_1^{\mathrm{inp}}, \alpha_2^{\mathrm{inp}}, ...]$ to produce a distribution over the input encodings, $\beta^{\mathrm{inp}} = \mathrm{softmax}\left(\alpha^{\mathrm{inp}}\right)$. The input representation $\kappa^{\mathrm{agg}}$ is the sum over the input encodings $h^{\mathrm{enc}}$ weighted by the normalized scores $\beta^{\mathrm{inp}}$:

$$\kappa^{\mathrm{agg}} = \sum_t \beta_t^{\mathrm{inp}} h_t^{\mathrm{enc}} \tag{3}$$

Let $\alpha^{\text{agg}}$ denote the scores over the aggregation operations such as COUNT, MIN, MAX, and the no-aggregation operation NULL. We compute $\alpha^{\text{agg}}$ by applying a multi-layer perceptron to the input representation $\kappa^{\text{agg}}$:

$$\alpha^{\text{agg}} = W^{\text{agg}} \tanh\left(V^{\text{agg}} \kappa^{\text{agg}} + b^{\text{agg}}\right) + c^{\text{agg}} \tag{4}$$

We apply the softmax function to obtain the distribution over the set of possible aggregation operations $\beta^{\text{agg}} = \text{softmax}\left(\alpha^{\text{agg}}\right)$. We use cross entropy loss $L^{\text{agg}}$ for the aggregation operation.

**SELECT Column.** The selection column depends on the table columns as well as the question. Namely, for the example in Figure 3, "How many engine types" indicates that we need to retrieve the "Engine" column. SELECT column prediction is then a matching problem, solvable using a pointer: given the list of column representations and a question representation, we select the column that best matches the question.

In order to produce the representations for the columns, we first encode each column name with a LSTM. The representation of a particular column $j$, $e_j^{\text{c}}$, is given by:

$$h_{j,t}^{\text{c}} = \text{LSTM}\left(\text{emb}\left(x_{j,t}^{\text{c}}\right), h_{j,t-1}^{\text{c}}\right) \qquad e_j^{\text{c}} = h_{j,T_j}^{\text{c}} \tag{5}$$

Here, $h_{j,t}^{\text{c}}$ denotes the $t$th encoder state of the $j$th column. We take the last encoder state to be $e_j^{\text{c}}$, column $j$'s representation.

To construct a representation for the question, we compute another input representation $\kappa^{\text{sel}}$ using the same architecture as for $\kappa^{\text{agg}}$ (Equation 3) but with untied weights. Finally, we apply a multi-layer perceptron over the column representations, conditioned on the input representation, to compute the a score for each column $j$:

$$\alpha_j^{\text{sel}} = W^{\text{sel}} \tanh\left(V^{\text{sel}} \kappa^{\text{sel}} + V^{\text{c}} e_j^{\text{c}}\right) \tag{6}$$

We normalize the scores with a softmax function to produce a distribution over the possible SELECT columns $\beta^{\text{sel}} = \text{softmax}\left(\alpha^{\text{sel}}\right)$. For the example shown in Figure 3, the distribution is over the columns "Entrant", "Constructor", "Chassis", "Engine", "No", and the ground truth SELECT column "Driver". We train the SELECT network using cross entropy loss $L^{\text{sel}}$.

**WHERE Clause.** We can train the WHERE clause using a pointer decoder similar to that described in Section 2.1. However, there is a limitation in using the cross entropy loss to optimize the network: the WHERE conditions of a query can be swapped and the query yield the same result. Suppose we have the question "which men are older than 18" and the queries SELECT name FROM insurance WHERE age > 18 AND gender = "male" and SELECT name FROM insurance WHERE gender = "male" AND age > 18. Both queries obtain the correct execution result despite not having exact string match. If the former is provided as the ground truth, using cross entropy loss to supervise the generation would then wrongly penalize the latter. To address this problem, we apply reinforcement learning to learn a policy to directly optimize the expected correctness of the execution result (Equation 7).

Instead of teacher forcing at each step of query generation, we sample from the output distribution to obtain the next token. At the end of the generation procedure, we execute the generated SQL query against the database to obtain a reward. Let $y = [y^1, y^2, ..., y^T]$ denote the sequence of generated tokens in the WHERE clause. Let $q(y)$ denote the query generated by the model and $q_g$ denote the ground truth query corresponding to the question. We define the reward $R(q(y), q_g)$ as

$$R(q(y), q_g) = \begin{cases} -2, & \text{if } q(y) \text{ is not a valid SQL query} \\ -1, & \text{if } q(y) \text{ is a valid SQL query and executes to an incorrect result} \\ +1, & \text{if } q(y) \text{ is a valid SQL query and executes to the correct result} \end{cases} \tag{7}$$

The loss, $L^{\text{whe}} = -\mathbb{E}_y[R(q(y), q_g)]$, is the negative expected reward over possible WHERE clauses. We derive the policy gradient for $L^{\text{whe}}$ as shown by Sutton et al. (2000) and Schulman et al. (2015).

$$\nabla L_\Theta^{\mathrm{whe}} \quad = \quad -\nabla_\Theta \left( \mathbb{E}_{y \sim p_y} \left[ R \left( q \left( y \right), q_g \right) \right] \right) \tag{8}$$

$$= \quad -\mathbb{E}_{y \sim p_y} \left[ R \left( q \left( y \right), q_g \right) \nabla_\Theta \sum_t \left( \log p_y \left( y_t; \Theta \right) \right) \right] \tag{9}$$

$$\approx \quad -R \left( q \left( y \right), q_g \right) \nabla_\Theta \sum_t \left( \log p_y \left( y_t; \Theta \right) \right) \tag{10}$$

Here, $p_y(y_t)$ denotes the probability of choosing token $y_t$ during time step $t$. In equation 10, we approximate the expected gradient using a single Monte-Carlo sample $y$

**Mixed Objective Function.** We train the model using gradient descent to minimize the objective function $L = L^{\mathrm{agg}} + L^{\mathrm{sel}} + L^{\mathrm{whe}}$. Consequently, the total gradient is the equally weighted sum of the gradients from the cross entropy loss in predicting the `SELECT` column, from the cross entropy loss in predicting the aggregation operation, and from policy learning.

## 3 WIKISQL

WikiSQL is a collection of questions, corresponding SQL queries, and SQL tables. A single example in WikiSQL, shown in Figure 2, contains a table, a SQL query, and the natural language question corresponding to the SQL query. Table 1 shows how WikiSQL compares to related datasets. Namely, WikiSQL is the largest hand-annotated semantic parsing dataset to date - it is an order of magnitude larger than other datasets that have logical forms, either in terms of the number of examples or the number of tables. The queries in WikiSQL span over a large number of tables and hence presents an

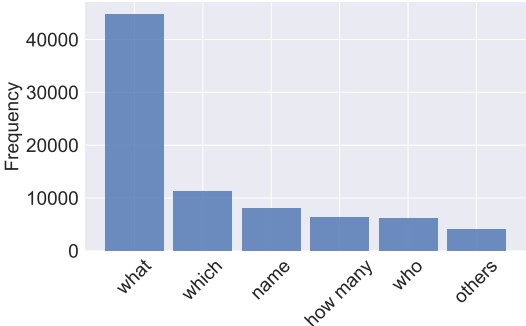

Figure 4: Distribution of questions in WikiSQL.

unique challenge: the model must be able to not only generalize to new queries, but to new table schema. Finally, WikiSQL contains realistic data extracted from the web. This is evident in the distributions of the number of columns, the lengths of questions, and the length of queries, respectively shown in Figure 5. Another indicator of the variety of questions in the dataset is the distribution of question types, shown in Figure 4.

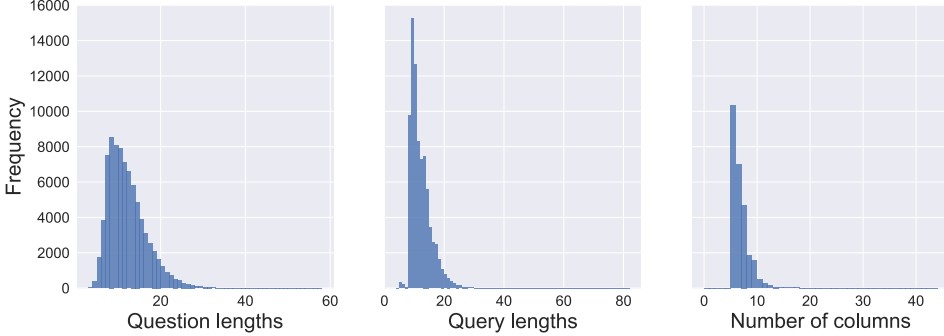

Figure 5: Distribution of table, question, query sizes in WikiSQL.

We collect WikiSQL by crowd-sourcing on Amazon Mechanical Turk in two phases. First, a worker paraphrases a generated question for a table. We form the generated question using a template, filled using a randomly generated SQL query. We ensure the validity and complexity of the tables by keeping only those that are legitimate database tables and sufficiently large in the number of rows and columns. Next, two other workers verify that the paraphrase has the same meaning as the generated question. We discard paraphrases that do not show enough variation, as measured by the character edit distance from the generated question, as well as those both workers deemed incorrect during verification. Section A of the Appendix contains more details on the collection

of WikiSQL. We make available examples of the interface used during the paraphrase phase and during the verification phase in the supplementary materials. The dataset is available for download at [MASK].

The tables, their paraphrases, and SQL queries are randomly slotted into train, dev, and test splits, such that each table is present in exactly one split. In addition to the raw tables, queries, results, and natural utterances, we also release a corresponding SQL database and query execution engine.

| Dataset | Size | LF | Schema |
|---|---|---|---|
| **WikiSQL** | **80654** | **yes** | **24241** |
| Geoquery | 880 | yes | 8 |
| ATIS | 5871 | yes* | 141 |
| Freebase917 | 917 | yes | 81* |
| Overnight | 26098 | yes | 8 |
| WebQuestions | 5810 | no | 2420 |
| WikiTableQuestions | 22033 | no | 2108 |

Table 1: Comparison between WikiSQL and existing datasets. The datasets are GeoQuery880 (Tang & Mooney, 2001), ATIS (Price, 1990), Free917 (Cai & Yates, 2013), Overnight (Wang et al., 2015), WebQuestions (Berant et al., 2013), and WikiTableQuestions (Pasupat & Liang, 2015). "Size" denotes the number of examples in the dataset. "LF" indicates whether it has annotated logical forms. "Schema" denotes the number of tables. ATIS is presented as a slot filling task. Each Freebase API page is counted as a separate domain.

### 3.1 EVALUATION

Let $N$ denote the total number of examples in the dataset, $N_{ex}$ the number of queries that, when executed, result in the correct result, and $N_{lf}$ the number of queries has exact string match with the ground truth query used to collect the paraphrase. We evaluate using the execution accuracy metric $Acc_{ex} = \frac{N_{ex}}{N}$. One downside of $Acc_{ex}$ is that it is possible to construct a SQL query that does not correspond to the question but nevertheless obtains the same result. For example, the two queries `SELECT COUNT(name) WHERE SSN = 123` and `SELECT COUNT(SSN) WHERE SSN = 123` produce the same result if no two people with different names share the SSN 123. Hence, we also use the logical form accuracy $Acc_{lf} = \frac{N_{lf}}{N}$. However, as we showed in Section 2.2, $Acc_{lf}$ incorrectly penalizes queries that achieve the correct result but do not have exact string match with the ground truth query. Due to these observations, we use both metrics to evaluate the models.

## 4 EXPERIMENTS

We tokenize the dataset using Stanford CoreNLP (Manning et al., 2014). We use the normalized tokens for training and revert into original gloss before outputting the query so that generated queries are executable on the database. We use fixed GloVe word embeddings (Pennington et al., 2014) and character n-gram embeddings (Hashimoto et al., 2016). Let $w_x^g$ denote the GloVe embedding and $w_x^c$ the character embedding for word $x$. Here, $w_x^c$ is the mean of the embeddings of all the character n-grams in $x$. For words that have neither word nor character embeddings, we assign the zero vector. All networks are run for a maximum of 300 epochs with early stopping on dev split execution accuracy. We train using ADAM (Kingma & Ba, 2014) and regularize using dropout (Srivastava et al., 2014). All recurrent layers have a hidden size of 200 units and are followed by a dropout of 0.3. We implement all models using PyTorch [1]. To train Seq2SQL, we first train a version in which the `WHERE` clause is supervised via teacher forcing (i.e. the policy is not learned from scratch) and then continue training using reinforcement learning. In order to obtain the rewards described in Section 2.2, we use the query execution engine described in Section 3.

### 4.1 RESULT

We compare results against the attentional sequence to sequence neural semantic parser proposed by Dong & Lapata (2016). This model achieves state of the art results on a variety of semantic parsing datasets, outperforming a host of non-neural semantic parsers despite not using hand-engineered grammars. To make this baseline even more competitive on our new dataset, we augment their input with the table schema such that the model can generalize to new tables. We describe this baseline in detail in Section 2 of the Appendix. Table 2 compares the performance of the three models.

---

[1] https://pytorch.org

| Model | Dev Acc$_{lf}$ | Dev Acc$_{ex}$ | Test Acc$_{lf}$ | Test Acc$_{ex}$ |
|---|---|---|---|---|
| Baseline (Dong & Lapata, 2016) | 23.3% | 37.0% | 23.4% | 35.9% |
| Aug Ptr Network | 44.1% | 53.8% | 43.3% | 53.3% |
| Seq2SQL (no RL) | 48.2% | 58.1% | 47.4% | 57.1% |
| **Seq2SQL** | **49.5%** | **60.8%** | **48.3%** | **59.4%** |

Table 2: Performance on WikiSQL. Both metrics are defined in Section 3.1. For Seq2SQL (no RL), the WHERE clause is supervised via teacher forcing as opposed to reinforcement learning.

Reducing the output space by utilizing the augmented pointer network improves upon the baseline by 17.4%. Leveraging the structure of SQL queries leads to another improvement of 3.8%, as is shown by the performance of Seq2SQL without RL compared to the augmented pointer network. Finally, training using reinforcement learning based on rewards from in-the-loop query executions on a database leads to another performance increase of 2.3%, as is shown by the performance of the full Seq2SQL model.

## 4.2 ANALYSIS

**Limiting the output space via pointer network leads to more accurate conditions.** Compared to the baseline, the augmented pointer network generates higher quality WHERE clause. For example, for "in how many districts was a successor seated on march 4, 1850?", the baseline generates the condition `successor seated = seated march 4` whereas Seq2SQL generates `successor seated = seated march 4 1850`. Similarly, for "what's doug battaglia's pick number?", the baseline generates `Player = doug` whereas Seq2SQL generates `Player = doug battaglia`. The conditions tend to contain rare words (e.g. "1850"), but the baseline is inclined to produce common words in the training corpus, such as "march" and "4" for date, or "doug" for name. The pointer is less affected since it selects exclusively from the input.

**Incorporating structure reduces invalid queries.** Seq2SQL without RL directly predicts selection and aggregation and reduces invalid SQL queries generated from 7.9% to 4.8%. A large quantity of invalid queries result from column names – the generated query refers to selection columns that are not present in the table. This is particularly helpful when

| Model | Precision | Recall | F1 |
|---|---|---|---|
| Aug Ptr Network | 66.3% | 64.4% | 65.4% |
| Seq2SQL | 72.6% | 66.2% | 69.2% |

Table 3: Performance on the COUNT operator.

the column name contain many tokens, such as "Miles (km)", which has 4 tokens. Introducing a classifier for the aggregation also reduces the error rate. Table 3 shows that adding the aggregation classifier improves the precision, recall, and F1 for predicting the COUNT operator. For more queries produced by the different models, please see Section 3 of the Appendix.

**RL generates higher quality WHERE clause that are ordered differently than ground truth.** Training with policy-based RL obtains correct results in which the order of conditions is differs from the ground truth query. For example, for "in what district was the democratic candidate first elected in 1992?", the ground truth conditions are `First elected = 1992 AND Party = Democratic` whereas Seq2SQL generates `Party = Democratic AND First elected = 1992`. When Seq2SQL is correct and Seq2SQL without RL is not, the latter tends to produce an incorrect WHERE clause. For example, for the rather complex question "what is the race name of the 12th round trenton, new jersey race where a.j. foyt had the pole position?", Seq2SQL trained without RL generates `WHERE rnd = 12 and track = a.j. foyt AND pole position = a.j. foyt` whereas Seq2SQL trained with RL correctly generates `WHERE rnd = 12 AND pole position = a.j. foyt`.

## 5 RELATED WORK

**Semantic Parsing.** In semantic parsing for question answering (QA), natural language questions are parsed into logical forms that are then executed on a knowledge graph (Zelle & Mooney, 1996; Wong & Mooney, 2007; Zettlemoyer & Collins, 2005; 2007). Other works in semantic parsing focus on learning parsers without relying on annotated logical forms by leveraging conversational

logs (Artzi & Zettlemoyer, 2011), demonstrations (Artzi & Zettlemoyer, 2013), distant supervision (Cai & Yates, 2013; Reddy et al., 2014), and question-answer pairs (Liang et al., 2011). Semantic parsing systems are typically constrained to a single schema and require hand-curated grammars to perform well[2]. Pasupat & Liang (2015) addresses the single-schema limitation by proposing the floating parser, which generalizes to unseen web tables on the WikiTableQuestions task. Our approach is similar in that it generalizes to new table schema. However, we do not require access to table content, conversion of table to an additional graph, hand-engineered features, nor hand-engineered grammar.

**Semantic parsing datasets.** Previous semantic parsing systems were designed to answer complex and compositional questions over closed-domain, fixed-schema datasets such as GeoQuery (Tang & Mooney, 2001) and ATIS (Price, 1990). Researchers also investigated QA over subsets of large-scale knowledge graphs such as DBPedia (Starc & Mladenic, 2017) and Freebase (Cai & Yates, 2013; Berant et al., 2013). The dataset "Overnight" (Wang et al., 2015) uses a similar crowd-sourcing process to build a dataset of natural language question, logical form pairs, but has only 8 domains. WikiTableQuestions (Pasupat & Liang, 2015) is a collection of question and answers, also over a large quantity of tables extracted from Wikipedia. However, it does not provide logical forms whereas WikiSQL does. WikiTableQuestions focuses on the task of QA over noisy web tables, whereas WikiSQL focuses on generating SQL queries for questions over relational database tables. We intend to build a natural language interface for databases, and do not use table content apart from evaluation.

**Representation learning for sequence generation.** Dong & Lapata (2016)'s attentional sequence to sequence neural semantic parser, which we use as the baseline, achieves state-of-the-art results on a variety of semantic parsing datasets despite not utilizing hand-engineered grammar. Unlike their model, Seq2SQL uses pointer based generation akin to Vinyals et al. (2015) to achieve higher performance, especially in generating queries with rare words and column names. Pointer models have also been successfully applied to tasks such as language modeling (Merity et al., 2017), summarization (Gu et al., 2016), combinatorial optimization (Bello et al., 2017), and question answering (Seo et al., 2017; Xiong et al., 2017). Another interesting neural semantic parsing model is the Neural Programmer by Neelakantan et al. (2017). Our approach is different than their work in that we do not require access to the table content during inference, which may be unavailable due to privacy concerns. We also do not hand-engineer model architecture for query execution and instead leverage existing database engines to perform efficient query execution. In contrast to both Dong & Lapata (2016) and Neelakantan et al. (2017), we train our model using policy-based RL, which helps Seq2SQL achieve state-of-the-art performance.

**Natural language interface for databases.** One of the prominent works in natural language interfaces is PRECISE (Popescu et al., 2003), which translates questions to SQL queries and identifies questions that it is not confident about. Giordani & Moschitti (2012) translate questions to SQL by first generating candidate queries from a grammar then ranking them using tree kernels. Both of these approaches rely on high quality grammar and are not suitable for tasks that require generalization to new schema. Iyer et al. (2017) also translate to SQL, but with a Seq2Seq model that is further improved with human feedback. Seq2SQL outperforms Seq2Seq and uses reinforcement learning instead of human feedback during training.

## 6 CONCLUSION

We proposed Seq2SQL, a deep neural network for translating questions to SQL queries. Our model leverages the structure of SQL queries to reduce the output space of the model. To train Seq2SQL, we applied in-the-loop query execution to learn a policy for generating the conditions of the SQL query, which is unordered and unsuitable for optimization via cross entropy loss. We also introduced WikiSQL, a dataset of questions and SQL queries that is an order of magnitude larger than comparable datasets. Finally, we showed that Seq2SQL outperforms a state-of-the-art semantic parser on WikiSQL, improving execution accuracy from 35.9% to 59.4% and logical form accuracy from 23.4% to 48.3%.

---

[2]For simplicity, we define table schema as the names of the columns in the table.

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

## A    COLLECTION OF WIKISQL

WikiSQL is collected in a paraphrase phases as well as a verification phase. In the paraphrase phase, we use tables extracted from Wikipedia by Bhagavatula et al. (2013) and remove small tables according to the following criteria:

- the number of cells in each row is not the same

- the content in a cell exceed 50 characters

- a header cell is empty

- the table has less than 5 rows or 5 columns

- over 40% of the cells of a row contain identical content

We also remove the last row of a table because a large quantity of HTML tables tend to have summary statistics in the last row, and hence the last row does not adhere to the table schema defined by the header row.

For each of the table that passes the above criteria, we randomly generate 6 SQL queries according to the following rules:

- the query follows the format `SELECT agg_op agg_col from table where cond1_col cond1_op cond1 AND cond2_col cond2_op cond2 ...`

- the aggregation operator `agg_op` can be empty or `COUNT`. In the event that the aggregation column `agg_col` is numeric, `agg_op` can additionally be one of `MAX` and `MIN`

- the condition operator `cond_op` is =. In the event that the corresponding condition column `cond_col` is numeric, `cond_op` can additionally be one of > and <

- the condition `cond` can be any possible value present in the table under the corresponding `cond_col`. In the event that `cond_col` is numerical, `cond` can be any numerical value sampled from the range from the minimum value in the column to the maximum value in the column.

We only generate queries that produce a non-empty result set. To enforce succinct queries, we remove conditions from the generated queries if doing so does not change the execution result.

For each query, we generate a crude question using a template and obtain a human paraphrase via crowdsourcing on Amazon Mechanical Turk. In each Amazon Mechanical Turk HIT, a worker is shown the first 4 rows of the table as well as its generated questions and asked to paraphrase each question.

After obtaining natural language utterances from the paraphrase phase, we give each question-paraphrase pair to two other workers in the verification phase to verify that the paraphrase and the original question contain the same meaning.

We then filter the initial collection of paraphrases using the following criteria:

- the paraphrase must be deemed correct by at least one worker during the verification phrase

- the paraphrase must be sufficiently different from the generated question, with a character-level edit distance greater than 10

## B    ATTENTIONAL SEQ2SEQ NEURAL SEMANTIC PARSER BASELINE

We employ the attentional sequence to sequence model for the baseline. This model by Dong & Lapata (2016) achieves state of the art results on a variety of semantic parsing datasets despite not using hand-engineered grammar. We implement a variant using OpenNMT and a global attention encoder-decoder architecture (with input feeding) described by Luong et al.

We use the same two-layer, bidirectional, stacked LSTM encoder as described previously. The decoder is almost identical to that described by Equation 2 of the paper, with the sole difference coming from input feeding.

$$g_s = \text{LSTM}\left(\left[\text{emb}\left(y_{s-1}\right); \kappa_{s-1}^{\text{dec}}\right], g_{s-1}\right) \tag{11}$$

where $\kappa_s^{\text{dec}}$ is the attentional context over the input sequence during the $s$th decoding step, computed as

$$\alpha_{s,t}^{\text{dec}} = h_s^{\text{dec}}\left(W^{\text{dec}}h_t^{\text{enc}}\right)^{\intercal} \qquad \beta_s^{\text{dec}} = \text{softmax}\left(\alpha_s^{\text{dec}}\right) \tag{12}$$

$$\kappa_s = \sum_t \beta_{s,t} h_t^{\text{enc}} \tag{13}$$

To produce the output token during the $s$th decoder step, the concatenation of the decoder state and the attention context is given to a final linear layer to produce a distribution $\alpha^{\text{dec}}$ over words in the target vocabulary

$$\alpha^{\text{dec}} = \text{softmax}\left(U^{\text{dec}}[h_s^{\text{dec}}; \kappa_s^{\text{dec}}]\right) \tag{14}$$

During training, teacher forcing is used. During inference, a beam size of 5 is used and generated unknown words are replaced by the input words with the highest attention weight.

## C  PREDICTIONS BY SEQ2SQL

| | |
|---|---|
| Q | when connecticut & villanova are the regular season winner how many tournament venues (city) are there? |
| P | SELECT COUNT tournament player (city) WHERE regular season winner city ) = connecticut & villanova |
| S' | SELECT COUNT tournament venue (city) WHERE tournament winner = connecticut & villanova |
| S | SELECT COUNT tournament venue (city) WHERE regular season winner = connecticut & villanova |
| G | SELECT COUNT tournament venue (city) WHERE regular season winner = connecticut & villanova |
| Q | what are the aggregate scores of those races where the first leg results are 0-1? |
| P | SELECT aggregate WHERE 1st . = 0-1 |
| S' | SELECT COUNT agg.  score WHERE 1st leg = 0-1 |
| S | SELECT agg.  score WHERE 1st leg = 0-1 |
| G | SELECT agg.  score WHERE 1st leg = 0-1 |
| Q | what is the race name of the 12th round trenton, new jersey race where a.j. foyt had the pole position? |
| P | SELECT race name WHERE location = 12th AND round position = a.j.  foyt, new jersey AND |
| S' | SELECT race name WHERE rnd = 12 AND track = a.j.  foyt AND pole position = a.j.  foyt |
| S | SELECT race name WHERE rnd = 12 AND pole position = a.j.  foyt |
| G | SELECT race name WHERE rnd = 12 AND pole position = a.j.  foyt |
| Q | what city is on 89.9? |
| P | SELECT city WHERE frequency = 89.9 |
| S' | SELECT city of license WHERE frequency = 89.9 |
| S | SELECT city of license WHERE frequency = 89.9 |
| G | SELECT city of license WHERE frequency = 89.9 |
| Q | how many voters from the bronx voted for the socialist party? |
| P | SELECT MIN % party = socialist |
| S' | SELECT COUNT the bronx where the bronx = socialist |
| S | SELECT COUNT the bronx WHERE the bronx = socialist |
| G | SELECT the bronx WHERE party = socialist |
| Q | in what year did a plymouth vehicle win on february 9 ? |
| P | SELECT MIN year (km) WHERE date = february 9 AND race time = plymouth 9 |
| S' | SELECT year (km) WHERE date = plymouth 9 AND race time = february 9 |
| S | SELECT year (km) WHERE date = plymouth 9 AND race time= february 9 |
| G | SELECT year (km) WHERE manufacturer = plymouth AND date = february 9 |

Table 4: Examples predictions by the models on the dev split. Q denotes the natural language question and G denotes the corresponding ground truth query. P, S', and S denote, respectively, the queries produced by the Augmented Pointer Network, Seq2SQL without reinforcement learning, Seq2SQL. We omit the FROM table part of the query for succinctness.

