# OpenReview forum: "Seq2SQL: Generating Structured Queries From Natural Language Using Reinforcement Learning "
_ICLR.cc/2018/Conference — Reject_

### Official Review · AnonReviewer3 · 2017-11-27
**This is a decent work but contains certain obvious drawbacks**

**Rating:** 5
**Confidence:** 5

**Review:**

This paper presents a new approach to support the conversion from natural language to database queries.

One of the major contributions of the work is the introduction of a new real-world benchmark dataset based on questions over Wikipedia. The scale of the data set is significantly larger than any existing ones. However, from the technical perspective, the reviewer feels this work has limited novelty and does not advance the research frontier by much. The detailed comments are listed below.

1) Limitation of the dataset: While the authors claim this is a general approach to support seq2sql, their dataset only covers simple queries in form of aggregate-where-select structure. Therefore, their proposed approach is actually an advanced version of template filling, which considers the expression/predicate for one of the three operators at a time, e.g., (Giordani and Moschitti, 2012).

2) Limitation of generalization: Since the design of the algorithms is purely based on their own WikiSQL dataset, the reviewer doubts if their approach could be generalized to handle more complicated SQL queries, e.g., (Li and Jagadish, 2014). The high complexity of real-world SQL stems from the challenges on the appropriate connections between tables with primary/foreign keys and recursive/nested queries.

3) Comparisons to existing approaches: Since it is a template-based approach in nature, the author should shrink the problem scope in their abstract/introduction and compare against existing template approaches. While there are tons of semantic parsing works, which grow exponentially fast in last two years, these works are actually handling more general problems than this submission does. It thus makes sense when the performance of semantic parsing approaches on a constrained domain, such as WikiSQL, is not comparable to the proposal in this submission. However, that only proves their method is fully optimized for their own template.

As a conclusion, the reviewer believes the problem scope they solve is much smaller than their claim, which makes the submission slightly below the bar of ICLR. The authors must carefully consider how their proposed approach could be generalized to handle wider workload beyond their own WikiSQL dataset.

PS, After reading the comments on OpenReview, the reviewer feels recent studies, e.g., (Guu et al., ACL 2017), (Mou et al, ICML 2017) and (Yin et al., IJCAI 2016), deserve more discussions in the submission because they are strongly relevant and published on peer-reviewed conferences.

---

> ### Author Response · Authors · 2017-12-30
> **RE: This is a decent work but contains certain obvious drawbacks**
>
> Thank you for your comments.
> 1. We recognize that the queries in WikiSQL are simple. It is not our intention to supplant existing models for SQL generation from natural languages. Our intention is to tackle the problem of generalizing across tables, which we believe is a key barrier to using such systems in practice. Existing tasks in semantic parsing and natural language interfaces focused on generation queries from natural language with respect to a single table. Our task requires performing this on tables not seen during training. We argue that while WikiSQL is not as complex as existing datasets in its query complexity, it is more complex in its generalization task.
>
> 2. We could not find existing tasks that focus on generalization to unseen tables, but recognize that we may have missed existing work that the reviewer is aware of. We would be happy to apply our methods to such a task.
>
> 3. We agree that the existing semantic parsing approach we compare against is more general. Our intention is to introduce baselines for the WikiSQL task that generalizes to unseen tables. The baselines are tailored to the particular task of generating SQL queries, but range from general and unstructured (e.g. augmented pointer) to templated and structured (e.g. WikiSQL). In addition, like Guu et al, Mou et al, Yin et al, we use reinforcement learning as a means to address equivalent queries.

---

### Official Review · AnonReviewer1 · 2017-11-27
**Good dataset but problematic claims.**

**Rating:** 4
**Confidence:** 4

**Review:**

This work introduces a new semantic parsing dataset, which focuses on generating SQL from natural language. It also proposes a reinforcement-learning based model for this task.

First of all, I'd like to emphasize that the creation of a large scale semantic parsing dataset is fantastic, and it is a much appreciated contribution. However, I find its presentation problematic. It claims to supplant existing semantic parsing and language-to-SQL datasets, painting WikiSQL as a more challenging dataset overall. Given the massive simplifications to what is considered SQL in this dataset (no joins, no subqueries, minimal lexical grounding problem), I am reluctant to accept this claim without empirical evidence. For example, how well does the proposed model work when evaluated on an existing dataset containing full SQL queries, such as ATIS? That being said, I am sympathetic to making simplifications to a dataset for the sake of scalability, but it shouldn't be presented as representative of SQL.

On the modeling side, the role of reinforcement learning seems oddly central in the paper, even though though the added complexity is not well motivated. RL is typically needed when there are latent decisions that can affect the outcome in ways that are not known a priori. In this case, we know the reward is invariant to the ordering of the tokens in the WHERE clause. There are far simpler solutions that would achieve the same result, such as optimizing the marginal likelihood or even simply including all orderings as training examples. These should be included as baselines.

While the data contribution is great, the claims of the paper need to be revised.

---

> ### Author Response · Authors · 2017-12-30
> **RE: Good dataset but problematic claims.**
>
> Thank you for your comments.
>
> 1. It is not at all our intention to claim that WikiSQL supplants existing datasets. Our intended emphasis is that WikiSQL requires that models generalize to tables not seen during training. We are not aware of a semantic parsing dataset that 1. Provides logical forms 2. Requires generalization to unseen tables/schemas 3. Is based on realistic SQL tables in relational databases. We do recognize that WikiSQL, in its current state, contains only simple SELECT-AGGREGATE-WHERE queries. More complex queries contain, as you said, joins and subqueries. We will take this into account and elaborate on the generation of WikiSQL (which we placed into the appendix due to length considerations). In particular, we will explicitly emphasize the fact that WikiSQL does not contain subqueries nor joins.
>
> 2. We agree that reinforcement learning seems like a general and complex solution to a specific problem that can be solved in other ways. In fact, another submission to ICLR leverages this insight to incorporate structures into the model to do, say, set prediction of WHERE conditions (https://openreview.net/forum?id=SkYibHlRb&noteId=S12EyE1bz). We chose to use the RL approach as the baseline for WikiSQL because it is easy to generalize this approach to other forms of equivalent queries should we expand WikiSQL in the future. We also found that it is simple to implement in practice.
> We agree though that given the current state of WikiSQL, there are simpler approaches to tackle the WHERE clause ordering problem. We incorporated your suggestion of augmenting the training set with all permutations of the WHERE clause ordering. By doing this, we obtained 58.97% execution accuracy and 45.32% logical form accuracy on the test set with the Seq2SQL model without RL. The higher execution accuracy and lower logical form accuracy suggests that annotators were biased and tended to agree with the WHERE clause ordering presented to them in the paraphrasing task. Because we permute the ordering of the WHERE clause in training, the model does not see this bias during training and obtains worse logical form accuracy. With RL and augmented training set, we obtained 59.6% execution accuracy and 45.7% logical form accuracy.

---

### Official Review · AnonReviewer2 · 2017-11-28
**Interesting paper, but with limited experiments**

**Rating:** 5
**Confidence:** 4

**Review:**

The authors have addressed the problem of translating natural language queries to SQL queries. They proposed a deep neural network based solution which combines the attention based neural semantic parser and pointer networks. They also released a new dataset WikiSQL for the problem. The proposed method outperforms the existing semantic parsing baselines on WikiSQL dataset.

Pros:
1. The idea of using pointer networks for reducing search space of generated queries is interesting. Also, using extrinsic evaluation of generated queries handles the possibility of paraphrasing SQL queries.
2. A new dataset for the problem.
3. The experiments report a significant boost in the performance compared to the baseline. The ablation study is helpful for understanding the contribution of different component of the proposed method.

Cons:
1. It would have been better to see performance of the proposed method in other datasets (wherever possible). This is my main concern about the paper.
2. Extrinsic evaluation can slow down the overall training. Comparison of running times would have been helpful.
3. More details about training procedure (specifically for the RL part) would have been better.

---

> ### Author Response · Authors · 2017-12-30
> **RE: Interesting paper, but with limited experiments**
>
> Thank you for your comments.
>
> 1. We computed the run time of the model with RL and without RL. There is a subtlety regarding the runtime computation in that we run the evaluation during each batch, which inherently does database lookup (e.g. to calculate the execution accuracy). The result of evaluation is used as reward in the case of reinforcement learning. Because of this, using RL does not really add to the compute cost, apart from propagating the actual policy gradients because reward computation is always done as a part of evaluation. Taking this into account, the per-batch runtime over an epoch for the no-RL model took 0.2316 seconds on average with a standard deviation of 0.1037 seconds, whereas the RL model took 0.2627 seconds on average with a standard deviation of 0.1414 seconds.
>
> 2. Regarding your main concern (that we compare to other datasets), we are not aware of other datasets for natural language to SQL generation that requires generalization to new table schemas. For example, WikiSQL contains ~20k table schemas while other SQL generation tasks focus on a single table. As a result, we decided to compare our model against an existing state of the art semantic parser on our task instead. We would be happy to study the effect of our proposed method on other datasets.
>
> 3. Our RL model is initialized with the parameters from the best non-RL model. This RL model is trained with Adam with a learning rate of 1e-4 and a batch size of 100. We use an embedding size of 400 (300 for word embeddings, 100 for character ngrams) and a hidden state size of 200. Each BiLSTM has 2 layers. Please let us know if there are any particular points you would like us to elaborate on.

---

### Public Comment · (anonymous) · 2017-11-06
**Novelty?**

It's amazing how the authors neglect all previous studies that essentially have done the Seq2SQL thing at least 3 times (arXiv:1612.01197; arXiv:1612.02741; arXiv:1704.07926). It's also amazing how the authors neglect the NIPS reviews which have already pointed this out.

---

> ### Author Response · Authors · 2017-11-06
> **RE: Novelty**
>
> Hi Anonymous,
>
> Can you please clarify your comment "essentially have done the Seq2SQL thing"? We reference several semantic parsing papers that convert natural language questions to logical forms. Moreover, we reference works that apply semantic parsing or other neural models to tables. We can and will certainly add the recent work you listed to our paper, but perhaps the phrase "neglect all previous studies" is a bit harsh?
>
> I am not certain as to whether it is appropriate to mention anonymous NIPS reviews, but the main concern from our NIPS reviews (e.g. the only negative review) was that we do not compare to semantic parsing results. We have since rewritten the paper to clarify this point. Namely, our baseline is a state-of-the-art neural semantic parser by Dong et al., who demonstrated its effectiveness on four semantic parsing datasets. The particular review you mention was actually the most positive, with its conclusion being "the experiment part is solid and convincing" and "I believe release of the datasets will benefit research in this direction."
>
> Regarding your concern about novelty:
>
> Prior work, including those you cite (and which we will certainly add to our references), mainly focus on semantic parsing over knowledge graphs or synthetic datasets. For example, the first work you reference by Liang et al. works on WebQuestionsSP, which has under 5k examples. The second work you reference by Mou et al. uses a dataset by Yin et. al that contains 25k synthetic examples from a single schema (the Olympic games table). Finally, your third reference by Guu et al uses SCONE, which is a synthetic semantic parsing dataset over 14k examples and 3 domains.
>
> In contrast, WikiSQL (which this paper introduces) spans 80k examples and 24k schemas over real-world web tables - orders of magnitude larger than previous efforts. The number of schemas, in particular, poses a difficult generalization challenge. Moreover, WikiSQL contains natural language utterances annotated and verified by humans instead of generated templates.  One of the novelties of our approach (Seq2SQL) is that while we operate on SQL tables, we do not observe the content of the table. That is, the rewards our model observes comes from database execution (as oppose to self execution). This also makes policy learning more challenging, because an important part of the environment (e.g. table content) is not observed. This is distinct from prior work, including those you reference, that learn using table content. Our approach forces the model to learn purely from the question and the table schema. This enables our model to act as a thin and scalable natural language interface instead of a database engine because it does not need to see the database content.
>
> Finally, as an impartial means to gauge the impact of our work, despite not having been published, WikiSQL is already seeing adoption and Seq2SQL is already being used as a reference baseline by the community (including submissions by other groups to this conference).

---

> > ### Public Comment · (anonymous) · 2017-11-08
> > **Novelty?**
> >
> > The authors cited previous semantic parsing papers using seq2seq models, but ignored all previous reinforcement learning-based Seq2SQL papers. This has already been reminded of by previous conference reviewers, but is completely neglected again in the revision. It is hard to feel the authors' will for making that kind of revision, which (although will significantly diminish the novelty claimed by this paper) however is unavoidable if the authors want to make this paper scientifically sound.

---

> > > ### Author Response · Authors · 2017-11-08
> > > **RE: Novelty**
> > >
> > > We thank the anonymous reviewer for the feedback and respectfully disagree regarding the novelty of our work. We refer readers back to our earlier comment regarding how our contribution is distinct from prior art. Once again, we regret not citing the anonymous reviewer's prior work (which we believe, while important, is distinct from ours). To the anonymous reviewer, I emphasize that we are not maliciously ignoring your work. We focused our efforts on addressing the main concern of the only negative review, which was that it was unclear how our model compares to to existing semantic parsing models. We have since addressed this in the fashion described by my previous comment.

---

> > ### Public Comment · (anonymous) · 2017-11-08
> > **NIPS**
> >
> > Regarding the different conclusions drawn by this NIPS review and the other anonymous reviewer, perhaps the authors should consider the possibility that the other anonymous reviewer did not write the NIPS review in question? In any event, I find it disturbing, albeit slightly amusing, that one would bring out recent and anonymous (and private, nonetheless) NIPS reviews in public like this. The area chair should make note of this and consider whether it is appropriate.

---

> > > ### Comment · Area_Chair · 2017-12-27
> > > **Anonymous review**
> > >
> > > Yes, this is inappropriate to bring out. I will ask reviewers to ignore the fact of private NIPS comments in their reviews.
> > >
> > > However, I do think the resulting discussion on past work is relevant and should be considered. (And also note that some conferences (NIPS->AIStats) do share past negative reviews.)

---

### Author Response · Authors · 2017-11-07
**Errata**

- There is a typo we will fix in the analysis of the WHERE clause in section 4.2. The example question should be "which males" instead of "which men". It is impossible for the model to generate the word "men" because it is not in the question nor the schema.

---

### Public Comment · ~Florin_Brad1 · 2017-11-10
**Related work**

Neat work!
We have also released a paper detailing a corpus for language to SQL generation, it might be of interest to you https://arxiv.org/abs/1707.03172

---

### Decision · Program_Chairs · 2018-01-29
**ICLR 2018 Conference Acceptance Decision**

**Decision:**

Reject

**Comment:**

This paper introduces a new dataset and method for a "semantic parsing" problem of generating logical sql queries from text.  Reviews generally seemed to be very impressed by the dataset portion of the work saying "the creation of a large scale semantic parsing dataset is fantastic," but were less compelled by the modeling aspects that were introduced and by the empirical justification for the work.  In particular:

- Several reviewers pointed out that the use of RL in particularly this style felt like it was "unjustified", and that the authors should have used simpler baselines as a way of assessing the performance of the system, e.g. "There are far simpler solutions that would achieve the same result, such as optimizing the marginal likelihood or even simply including all orderings as training examples"

- The reviewers were not completely convinced that the authors' backed up their claims about the role of this dataset as a novel contribution. In particular there were questions about its structure, e.g. "dataset only covers simple queries in form of aggregate-where-select structure" and about comparisons with other smaller but similar datasets, e.g.  "how well does the proposed model work when evaluated on an existing dataset containing full SQL queries, such as ATIS"

There was an additional anonymous discussion about the work not citing previous semantic parsing datasets. The authors noted that this discussion inappropriately brought in previous private reviews. However it seems like the main reviewers issues were orthogonal to this point, and so it was not a major aspect of this decision.